# Evolution of Status of Trace Elements and Metallothioneins in Patients with COVID-19: Relationship with Clinical, Biochemical, and Inflammatory Parameters

**DOI:** 10.3390/metabo13080931

**Published:** 2023-08-09

**Authors:** Lourdes Herrera-Quintana, Héctor Vázquez-Lorente, Yenifer Gamarra-Morales, Jorge Molina-López, Elena Planells

**Affiliations:** 1Department of Physiology, School of Pharmacy, Institute of Nutrition and Food Technology “José Mataix”, University of Granada, 18071 Granada, Spain; elenamp@ugr.es; 2Clinical Analysis Unit, Valle de Los Pedroches Hospital, Pozoblanco, 14400 Córdoba, Spain; jennifer.gamarra.sspa@juntadeandalucia.es; 3Faculty of Education, Psychology and Sports Sciences, University of Huelva, 21007 Huelva, Spain; jorge.molina@ddi.uhu.es

**Keywords:** iron, zinc, copper, manganese, metallothionein, critical care, COVID-19

## Abstract

The inflammatory reaction and pathogenesis of COVID-19 may be modulated by circulating trace elements (Iron (Fe), Zinc (Zn), Copper (Cu), Manganese (Mn)) and Metallothioneins (MTs). Thus, the present study aimed to investigate their relationship with clinical, biochemical, and inflammatory parameters in patients with COVID-19 at the early Intensive Care Unit (ICU) phase. Critically ill patients (*n* = 86) were monitored from the first day of ICU admission until the third day of stay. Serum samples were used to assess mineral levels via Inductively Coupled Plasma Mass Spectrometry (ICP-MS) and MT levels via differential pulse voltammetry. Levels of Cu and MTs were significantly decreased after 3 days (*p* < 0.05), increasing the prevalence of Cu-deficient values from 50% to 65.3% (*p* = 0.015). Fe and Zn were shown to have a predictive value for mortality and severity. The present study suggests trace element deficiency may be a risk factor during early ICU treatment of COVID-19, as it is related to different biochemical and clinical parameters, indicating a possible beneficial effect of restoring proper levels of these micronutrients.

## 1. Introduction

Coronavirus Disease-19 (COVID-19) is a severe acute respiratory syndrome caused by the Severe Acute Respiratory Syndrome Coronavirus 2 (SARS-CoV-2), which rapidly spread throughout the world and has been declared a pandemic, with a need for hospitalization and intensive therapy in severe cases [1]. Apart from the well-known respiratory symptoms (e.g., respiratory distress, low oxygen saturation, or respiratory failure), all of them considered to be the main symptoms of COVID-19, the appearance of coagulopathy or cytokine storm processes are also clinical signs and symptoms involved in the systemic inflammatory process that occurs during COVID-19 [2,3]. Despite the improvement in disease management, most of the therapies have shown no perceivable impact, there is no existing universal cure for COVID-19 disease to date, and the rapidly mutating virus has emerged as a matter of great concern—the effort towards combating the disease should not be limited to finding drugs and vaccines [4,5]. In this regard, a proper nutritional status, which has an important role in optimal well-being and immune function, may be an interesting alternative [6]. Thus, some current therapies in clinical settings consist of immunomodulatory agents, including micronutrients, to fight against oxidative stress and modulate inflammation [7].

Trace elements (i.e., Iron (Fe), Zinc (Zn), Copper (Cu), and Manganese (Mn)) are micronutrients mostly responsible for modulating the inflammatory reaction and cytokine production in patients with COVID-19, and their deficiency is responsible for the increased risk of morbidity and mortality among these patients [8,9]. It must be noted that excess in circulating mineral levels may also be associated with different complications: (I) Fe overload, as a consequence of the decomposition of the hemoglobin by SARS-CoV-2, causes oxidative damage, inflammation, and immune dysfunction, which can lead to rapid multi-organ failures [10]; (II) excessive free Zn ions may be toxic by binding at the active site of enzymes or via the allosteric inhibition of enzymes [11], and high concentrations of Zn inhibit T cell functions [12]; (III) free Cu may catalyze the formation of highly reactive hydroxyl radicals [13]; and (IV) Mn at elevated levels is neurotoxic [14]. Thus, maintaining proper levels of these trace elements is crucial for a good state of health.

Metal ion concentrations in the body are controlled by the affinity for their ligands [15]. Metallothioneins (MTs) are a family of ubiquitous small proteins with a high cysteine content, which confers on them an optimal capacity for metal ion coordination via detoxification, storage, and delivery, participating in a wide range of stress responses, tumorigenesis, neurodegeneration, and inflammatory processes [16]. MTs bind various divalent trace metal ions, protecting cells and tissues against heavy metal toxicity and playing an essential role in intracellular aspects, especially for Zn and Cu [17]. Additionally, it has been shown that circulating MTs can be increased by different oxidative processes in multiple organ systems, although their production often decreases to protect against oxidative tissue injury [18]. Moreover, the overexpression and up-regulation of MTs have been associated with carcinogenic processes [17].

To date, the relationship between circulating levels of minerals and MTs and their subsequent effects on clinical and inflammatory parameters during COVID-19 has not been investigated. Therefore, the present study aimed to analyze the association between circulating Fe, Zn, Cu, Mn, and MT concentrations and their relationship with clinical, biochemical, and inflammatory parameters in critical care patients with COVID-19 at Intensive Care Unit (ICU) admission and after 3 days of ICU stay. We hypothesized that trace elements could be directly related to MT levels, worsening patients’ status during the ICU stay.

## 2. Materials and Methods

### 2.1. Subjects and Study Design

A prospective observational and analytical study was performed on 86 critically ill patients (18 women) aged 36–96 years with COVID-19, who were monitored from the first day of admission to the ICU (baseline) until the third day of stay (follow-up) at the Hospital Virgen de las Nieves in the province of Granada (Spain). The participants were recruited from 1st April to 1st December 2020 after being informed about the study protocol, which all the patients or their families signed. The diagnosis of COVID-19 was based on a positive Real-Time Reverse Transcriptase–PCR (RT-PCR) test and subsequent RNA sequencing specifically for the SARS-CoV-2 testing of nasal and pharyngeal swab samples. The participants were considered critically ill when they presented a respiratory failure requiring mechanical ventilation, needed vasopressor treatment (shock), or had other complications with organ failure requiring monitoring or treatment in the ICU. The inclusion criteria were as follows: (I) 18 years of age or older, (II) to have been previously hospitalized for at least more than 48 h, (III) to have been admitted to the ICU to stay for at least 3 days, and (IV) to present a positive PCR test for SARS-CoV-2 according to the Chinese Clinical Guideline for the classification of COVID-19 [19]. The exclusion criteria were as follows: (I) to be under 18 years, (II) to be pregnant, and (III) to not present a positive PCR test even though they presented symptoms compatible with COVID-19. The present study was conducted in accordance with the principles of the Declaration of Helsinki (last revised guidelines, 2013) [20], following the International Conference on Harmonization/Good Clinical Practice standards, and it was approved by the Ethics Committee of the University of Granada (Ref. 149/CEIH/2016).

### 2.2. Data Collection

Data, including positive PCR results against SARS-CoV-2, age, sex, exitus, ICU length of stay, and respiratory and clinical parameters, were retrieved from the hospital electronic database system and recorded for each participant on the first and third days of ICU admission, respectively. The respiratory and clinical parameters collected by the intensivists were: Sequential Organ Failure Assessment (SOFA) scale score [21], Acute Physiology and Chronic Health Evaluation II (APACHE II) scale score [22], days on mechanical ventilation (MV), Mean Arterial Pressure (MAP), Heart Rate (HR), Breath Rate (BR), Fraction of Inspired Oxygen (FiO_2_), Partial Oxygen arterial pressure/FiO_2_ (PaO_2_/FiO_2_), and Positive End-Expiratory Pressure (PEEP).

### 2.3. Blood Sampling and Biochemical Parameters Analysis

Blood samples were collected in the morning under fasting conditions, followed by centrifugation (4 °C for 15 min at 3500 rpm) to separate plasma and serum. The samples were frozen at −80.0 °C until the analysis of the different parameters. All measurements were obtained in triplicate, and blind quality control samples were included in the same assay batches to determine laboratory error in the measurements.

#### 2.3.1. Blood Sampling and Biochemical Parameters Analysis

The biochemical and inflammatory variables (i.e., albumin, ferritin, transferrin, Transferrin Saturation Index (TSI), fibrinogen, D-dimer, C-Reactive Protein (CRP), Glutamic Oxaloacetic Transaminase (GOT), Glutamic Pyruvic Transaminase (GPT), Gamma-Glutamyl Transferase (GGT), Lactate Dehydrogenase (LDH), Hemoglobin (Hb), leukocytes, percentage of lymphocytes and neutrophils, platelets, Troponin T (TNT), Activated Partial Thromboplastin Time (APTT), and International Normalized Ratio (INR)) were analyzed in the Virgen de las Nieves Hospital—which provided the reference values—using “Alinity,” the Abbott Core Laboratory^®^ autoanalyzer for biochemistry and immunochemistry, following enzymatic colorimetry and immunoassay procedures.

The mineral variables (i.e., Fe, Zn, Cu, and Mn) were analyzed using Inductively Coupled Plasma Mass Spectrometry (ICP-MS) (Perkin Elmer^®^ SCIEX Elan-5000 equipment, Markham, ON, Canada) following a previous wet-mineralized way in the Scientific Instrumentation Center from the University of Granada—which provided the reference values—as previously described [23].

#### 2.3.2. Sample Processing and Differential Pulse Voltammetry Brdicka Reaction for Determination of MTs

A total of 10.0 µL of serum samples was mixed with 990.0 µL of 0.1 M phosphate buffer (Na_2_HPO_4_ + NaH_2_PO_4_, pH 7.0) and subsequently incubated at 99.0 °C in a thermomixer (Eppendorf 5430, Hamburg, Germany) for 20 min with shaking to remove ballast proteins and peptides, which could influence the electrochemical response. Heat treatment effectively denatures and removes non-thermostable high-molecular-weight proteins from samples [24]. The denatured homogenates were centrifuged at 4 °C and 25,000× *g* for 20 min (Eppendorf 5402, Hamburg, Germany), and the obtained supernatant was used for MTs analysis.

Differential pulse voltammetry measurements for the determination of MTs in the Brdicka reaction according to the modified Brdicka procedure [17] were performed in the processed samples with a potentiostat/galvanostat AUTOLAB PGSTAT204 (Methrom, Utrecht, The Netherlands) connected to a VA-Stand 663 (Metrohm, Utrecht, The Netherlands), using a standard cell and 3 electrodes: (I) a hanging mercury drop electrode with a drop area of 0.4 mm^2^ was used as the working electrode, (II) a Ag/AgCl/3M KCl electrode was the reference, and (III) a platinum electrode was the auxiliary electrode. A total of 15 mL of a Brdicka supporting electrolyte containing 1 mM Co(NH_3_)_6_Cl_3_ and 1 M ammonia buffer (NH_3_(aq) + NH_4_Cl, pH 9.6) was used without surface-active agent additives and mixed with 5.0 μL of the processed samples. The Brdicka reaction of MTs follows the principle of the adsorptive stripping technique and is based on the strong adsorption of the target molecule on the surface of the working electrode at an open electrode circuit. After each measurement, the hanging mercury drop electrode and Brdicka supporting electrolyte are renewed. The measurement parameters were as follows: initial potential of −0.7 V, end potential of −1.8 V, modulation time of 0.05 s, time interval of 0.57 s, step potential of −3.9 mV, and modulation amplitude of 25.0 mV. All experiments were performed at 5 °C using the thermostat Julabo F25 (Labortechnik, Wasserburg, Germany). MTs were quantified from the calibration straight line using a commercially available ≥95%—pure solution isolated from rabbit liver (Enzo Life Sciences, Inc., New York, NY, USA). The obtained calibration curve for MTs was as follows y = 2.017x − 0.023; R^2^ = 0.996. NOVA 2.1.4 Metrohm Autolab B.V. (EcoChemie, Utrecht, The Netherlands) was used for data processing.

### 2.4. Statistical Analysis

The data’s normal distribution was examined using the Kolmogorov–Smirnov test prior to further statistical evaluation. Quantitative parameters were represented as the arithmetic mean (Standard Deviation (SD)). The paired Student’s *t*-test for parametric samples was used for the comparative analyses of all the clinical, biochemical, inflammatory, mineral variables, and MTs of the study. A Chi Square test was employed to compare the percentage of deficiency of all minerals between the 1st and 3rd days of ICU admission. The unpaired Student’s *t*-test for parametric samples was used for the comparative inter-group analysis (i.e., changes in clinical, biochemical, and inflammatory parameters of the study observed via mean changes in mineral and MT levels) after 3 days of ICU stay. To verify if mineral levels were predicting mortality and severity at the beginning and during follow-up, the Area Under the Curve (AUC) was evaluated using the Receiver Operating Curve (ROC). Correlation analyses and partial correlation coefficients between mean changes of all parameters of the study after 3 days of ICU stay were performed using Pearson’s correlation. The Benjamini–Hochberg (BH) procedure was employed to adjust *p*-Values, controlling the False Discovery Rate (FDR) [25]. The SPSS version 26.0 statistical package (IBM SPSS, Armonk, New York, NY, USA) was used throughout the study. Statistical significance was considered for *p* < 0.05. GraphPad Prism version 9.0 software (GraphPad Software, San Diego, CA, USA) was used for plotting the graphs.

## 3. Results

The clinical parameters of the study participants are represented in Appendix A. The BR, FiO_2_, and PEEP decreased after 3 days of ICU stay (all *p* ≤ 0.038), whereas the rest of the parameters showed no mean differences (all *p* ≥ 0.142) on the 3rd day of ICU stay in the total population. In general terms, the non-survivors presented a worst clinical profile with respect to the survivors when comparing with the reference values. After Benjamini–Hochberg correction, only the decrease in BR and FiO_2_ was significant.

Table 1 represents the biochemical, inflammatory, mineral, and metallothionein values of critically ill patients on the 1st and 3rd days of ICU stay. The fibrinogen, CRP, LDH, hemoglobin, hematocrit, percentage of neutrophils, Cu, and MTs decreased significantly (all *p* ≤ 0.046). In contrast, the D-dimer, GPT, GGT, percentage of lymphocytes, and platelets increased (all *p* ≤ 0.036) on the 3rd day of ICU stay. The rest of the parameters showed no mean differences after 3 days of ICU stay (all *p* ≥ 0.151). The mineral variables of the study and the MTs are additionally represented in Appendix A.

Figure 1 represents the percentage of patients with a deficient status in the studied minerals on the 1st and 3rd days of ICU stay. A relatively high prevalence of mineral deficiency (i.e., 11.9–50%) at ICU admission was observed, increasing (i.e., 14.3–65.3%) on the 3rd day of ICU stay—in the case of Cu being significant (*p* = 0.015). The deficiency prevalence in the case of the remaining minerals showed no significant differences after 3 days of ICU stay (all *p* ≥ 0.05).

Table 2 shows the differences in changes in the clinical, biochemical, and inflammatory parameters occurring during ICU stay based on the median changes in mineral and MT levels to check if higher differences were exhibited below or above the median values of the mentioned variables. Higher changes in Fe levels were found in patients presenting higher changes in HR (*p* = 0.005). Moreover, higher changes in TSI, fibrinogen, CRP, and neutrophils (all *p* ≤ 0.031) and lower changes in lymphocyte levels (*p* = 0.023) were found in those presenting higher changes in Cu levels. Furthermore, individuals with higher changes in Mn showed higher changes in albumin (*p* = 0.044) and lowered CRP (*p* = 0.038) levels. The rest of the variables showed no differences based on the median changes in the level of minerals and MTs (all *p* ≥ 0.073). After the Benjamini–Hochberg correction, these differences were no longer significant, except for changes in fibrinogen and changes in Cu levels (*p* = 0.027).

Table 3 represents the relationship between changes in minerals and MTs with clinical, biochemical, and inflammatory parameters after 3 days of evolution in the ICU. Changes in Fe were directly related to changes in Cu and Mn (all r ≥ 0.266; *p* ≤ 0.019). Changes in Zn were positively linked to changes in PaO_2_/FiO_2_ and platelets (all r ≥ 0.494; *p* ≤ 0.010) and negatively associated with changes in BR and INR (all r ≥ −0.480; *p* ≤ 0.026). Changes in Cu were positively related to changes in fibrinogen and CRP (all r ≥ −0.276; *p* ≤ 0.019). Changes in Mn were directly associated with changes in fibrinogen (r = 0.250; *p* = 0.039) and negatively related to changes in GOT (r = 0.221; *p* = 0.049). Finally, changes in MTs were inversely associated with changes in Mn and albumin (all r ≥ −0.255; *p* ≤ 0.039). The rest of the variables showed no relationship between them (all *p* ≥ 0.05). After the Benjamini–Hochberg correction, several differences were no longer significant.

Figure 2 shows the ROC curves of Fe and Zn as mortality and severity predictors. The AUC values were 0.655 (95% CI 0.533–0.778) for changes in Fe predicting mortality and 0.652 (95% CI 0.531–0.773) for Fe predicting SOFA at ICU admission. In the same way, the AUC values were 0.709 (95% CI 0.503–0.914) for Zn predicting APACHE on the 1st day of ICU stay and 0.768 (95% CI 0.586–0.951) for Zn predicting SOFA on the 3rd day of ICU stay. The ROC curves of the rest of minerals showed values of AUC around 0.5.

## 4. Discussion

The main findings of the present study showed a relatively high prevalence of mineral deficiency at ICU admission, increasing after 3 days. Additionally, a decrease in MT levels was observed during this period. Changes in Fe were a prognostic biomarker of severity and mortality, and changes in Zn levels were also predictors of severity. Contrary to our previous hypothesis, no relationship was found between changes in MTs and changes in both Zn and Cu levels. The present study highlights the potential benefit of assessing the mineral status and MTs at early stages in critical patients with severe COVID-19 infections to characterize the disease, identify and reverse deficiency conditions, and improve the treatment and prognosis of these patients.

SARSCoV-2 infection is a heterogeneous illness because not all patients develop the same symptoms. However, some parameters have been reported as a common finding in cases with worse prognosis (i.e., thrombocytopenia or higher CRP, D-dimer, transferrin, and LDH levels, among others) [26]. In our study, patients showed altered values for most clinical and biochemical parameters at ICU admission, and changes with an apparent trend towards normalized values (i.e., BR, PaO_2_/FiO_2_, fibrinogen) were observed after 3 days. However, levels of many variables remained altered, worsening during ICU stay (i.e., D-dimer, GGT, GPT), underscoring the complexity of the clinical situation of these patients. An over-exuberant inflammatory response and dysregulated host immune system are common findings in severe cases of COVID-19 [27]. Among other effects, inflammatory processes lead to oxidative stress and altered levels of different trace elements in serum [28].

Regarding Fe levels, our results showed normal mean values for this mineral at baseline and follow-up, with a prevalence of deficiency of around 10%. Moreover, Fe levels at baseline predicted severity, and changes in its levels during ICU stay predicted mortality. The relationship between Fe levels and susceptibility to infection remains complex because, on the one hand, Fe could reduce the vulnerability to respiratory infections via enhancing natural killer cell activity, T lymphocyte proliferation, or production of T helper 1 cytokines, but, on the other, infections caused by organisms that spend part of their life-cycle intracellularly may be enhanced by Fe [29]. Indeed, it has been described that coronavirus can capture Fe and generate reactive oxygen species to damage the human immune system [30]. Additionally, it must be noted that, although Fe levels were within normal values, important disturbances were found in ferritin levels. It has been reported that changes in Fe metabolism can be used to predict death in patients admitted to ICU because serum ferritin has been identified as one of the predictors of death in COVID-19 patients [31,32]. In addition, the development of anemia in critically ill patients is frequent—along with elevated ferritin levels—due to several factors, including Fe-withholding mechanisms caused by inflammation [33]. Hyperferritinemia—a key acute-phase reactant—could represent a host defense mechanism depriving bacterial growth of iron and protecting immune cell function. However, there is a need for further investigation of the role of ferritin and Fe metabolism in uncontrolled inflammatory conditions [31].

Similarly, we found normal mean values of circulating Zn—more than one-third of the patients presenting deficient levels—at ICU admission and after 3 days of stay. Lower Zn levels during the early phase of ICU stay in patients with COVID-19 infection have been reported previously [34,35]. Hypozincemia in these patients is characteristic, and it is possible due to redistribution processes from blood to the liver at the expense of Zn in other tissues [36]. Furthermore, we found that changes in Zn levels were directly associated with changes in PaO_2_/FiO_2_ and platelets and inversely related to changes in the BR and INR (although differences were no longer significant after the Benjamini–Hochberg correction). Likewise, Zn levels presented predictive values for severity scales. These results, altogether, appear to suggest that higher levels of Zn are related to better clinical outcomes. In consistency with that, Maares et al. [37] found an association between serum Zn levels and an increased risk of death, suggesting that Zn may serve as a prognostic marker for the severity and course of COVID-19. Additionally, Notz et al. [38] reported that circulating Zn—below the reference range in patients with severe COVID-19—was restored after supplementation within two weeks of intensive care, supporting the possible beneficial effect of Zn supplementation.

Circulating Cu levels decreased during ICU stay—the mean levels being below normal values—reaching 65% for the prevalence of deficiency during the first 3 days of ICU stay. Cu levels were directly related to fibrinogen, platelets, and CRP levels and inversely related to the INR (with significant differences only in the case of fibrinogen after the Benjamini–Hochberg correction). Moreover, patients with higher changes in Cu levels presented higher variations in TSI and neutrophils and lower changes in lymphocyte levels (although these associations were no longer significant after the Benjamini–Hochberg correction). A decrease in lymphocyte levels has been previously linked to a higher mortality risk, and an increase in CRP is linked to a severe prognosis in COVID-19 patients [26]. Our study shows that better clinical, biochemical, and inflammatory parameters are associated with lower concentrations of circulating Cu during this acute phase of infection. Indeed, Cu levels have been previously associated with CRP, suggesting an association between serum Cu levels and COVID-19 severity [34]. Thus, a possible inflammatory and pro-oxidant role of Cu in COVID-19 pathogenesis has been hypothesized [39]. In contrast, Hackler et al. [40] found elevated serum Cu levels at hospital admission in the surviving patients compared with non-survivors and with the reference range; however, no patient showed Cu levels below the reference range. These discrepancies could be due to differences in patients’ severity and inflammatory conditions. Further research is needed to know the role of circulating Cu on outcomes in critically ill patients with COVID-19.

Concerning Mn levels, our study revealed that the prevalence of deficiency affected one-third of patients at follow-up, with Mn changes inversely related to changes in GOT and MT levels and directly associated with changes in fibrinogen, Fe, and Cu (which were not significant after the Benjamini–Hochberg correction). No differences in Mn levels were found by Zhou et al. [41] when they compared mild and severe groups of patients with COVID-19. However, Zeng et al. [42] found higher urinary excretion of Mn and other trace elements in severe patients than in non-severe cases. In general, Mn levels appear to follow the trend of other trace elements, and Mn could play a role in the pathogenesis of this disease. However, the evidence assessing Mn levels and their effects on COVID-19 is scarce and inconclusive to date.

Although MTs play a major role in the homeostasis of Zn and Cu and detoxification processes, their biological function is still debatable [43,44]. In cellular processes, MTs are known to have metallo-regulatory functions (i.e., growth, differentiation, and protective role in oxidative stress) [17]. Our results showed a significant decrease in the concentration of circulating MTs during ICU stay, and these changes in MT levels are inversely correlated to changes in albumin levels (before the Benjamini–Hochberg correction). Metal-Cys clusters in MTs—together with disulfide moieties in serum albumins—are considered good interceptors of free radicals [45]. Therefore, circulating MTs could be determinant in antioxidant processes; their levels are influenced by oxidative stress and inflammation. Moreover, contrary to our previous hypothesis, no relationship was found between changes in MT levels and Zn and Cu levels; they are only inversely associated with changes in Mn levels (before the Benjamini–Hochberg correction). It has been observed that chronic low-level exposure to Mn in rats significantly decreases hepatic MTs—the liver being the organ with the highest concentration of MTs [46]. Furthermore, the effect of inflammation on circulating MTs and mineral levels remains unclear. In patients with chronic exacerbated pancreatitis, a significant increase in the concentration of circulating MTs was accompanied by a decrease in Zn and an increase in Cu levels [47]. Nevertheless, MT levels were lower in patients with hepatitis C than in healthy controls. In contrast, in other chronic hepatitis cases, MT levels increased gradually, followed by the progression of the disease to liver cirrhosis and hepatocellular carcinoma [48]. Further studies are needed to fully understand the relationships between MTs and mineral levels and their implications in clinical outcomes, especially during inflammatory conditions. Moreover, the last revised European Society of Parenteral and Enteral Nutrition (ESPEN) guidelines [49] about early nutritional supply and micronutrient supplements should also be considered in these patients’ management.

The findings reported in the present study should be treated with caution as some limitations arise. Firstly, follow-up during the 3-day stay in the ICU did not allow us to establish causal relationships as no intervention was performed. Secondly, the recruited patients were from a single hospital, and some potential confounding factors (sociodemographic and socioeconomic status) were not evaluated. Thus, these outcomes cannot be generalized to other populations, especially considering the wide range of COVID-19 prevalence. Thirdly, it was not possible to include critical patients without COVID-19 as a control group in order to compare the present findings. Fourthly, information regarding specifical medication usage (e.g., glucocorticoids as dexamethasone or similar medication) was not available. Finally, the overall negative results may be related to the heterogeneity of the subjects and their underlying disease conditions or severity. Replicating this study in a larger and heterogeneous critically ill population with a control group could further corroborate our findings.

## 5. Conclusions

In summary, the present study of critically ill patients with COVID-19 indicated an apparent risk of a deficiency of trace elements at ICU admission, worsening after 3 days of stay. Cu and MT levels decreased significantly during this period, whereas Fe and Zn levels were shown as mortality and severity predictors. Different biochemical and clinical parameters were related to the studied variables. Therefore, we suggest monitoring the mineral status and performing nutritional interventions, when appropriate, that could help to improve the altered parameters, such as inflammatory conditions, and, thus, the prognosis in critically ill patients with COVID-19.

## Figures and Tables

**Figure 1 metabolites-13-00931-f001:**
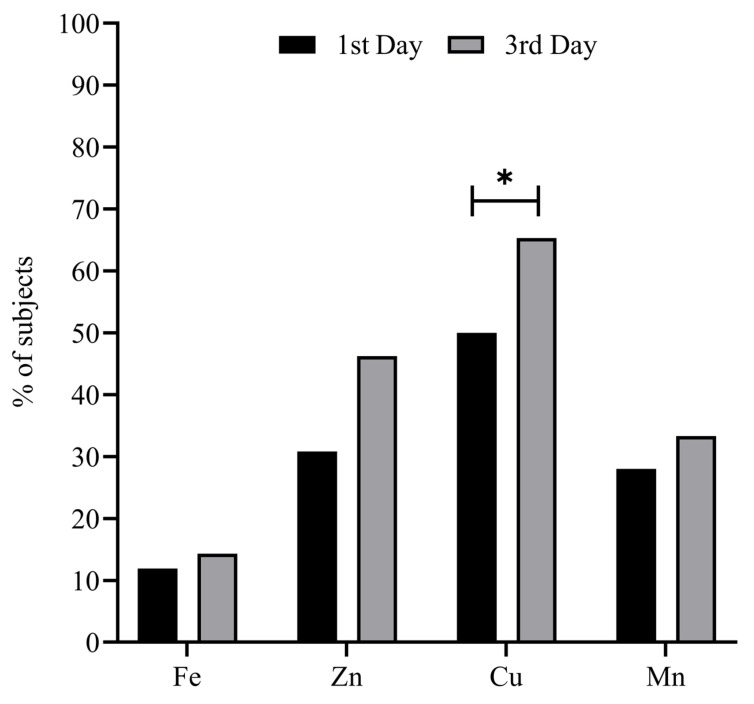
Percentage of subjects with mineral deficient status on the 1st and 3rd days of ICU stay. Chi Square test was used to compare the evolution of the mineral variables by % of deficient subjects. * Statistical significance = *p* < 0.05. Abbreviations: Cu = copper; Fe = iron; Mn = manganese; Zn = zinc.

**Figure 2 metabolites-13-00931-f002:**
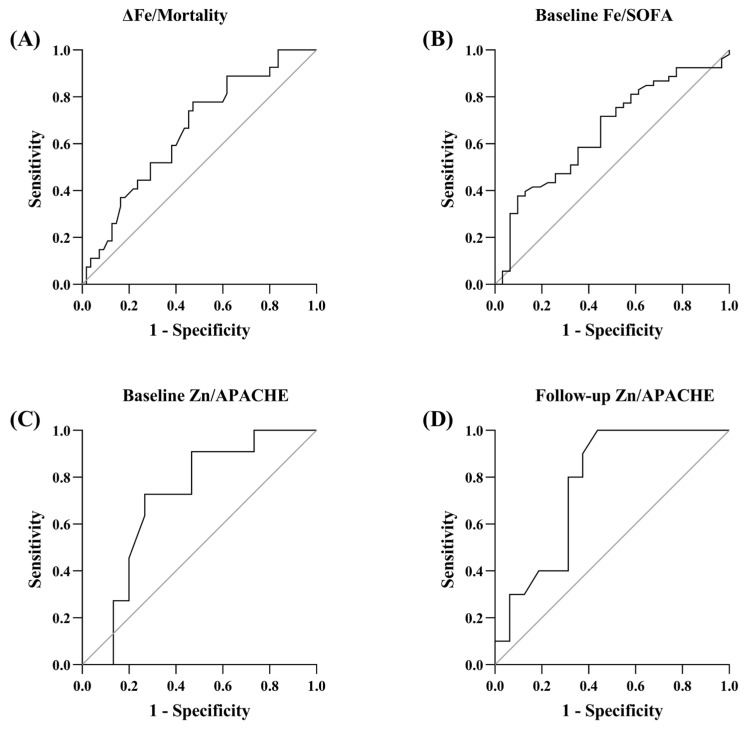
ROC curves of Fe and Zn as mortality and severity predictors. (**A**) ROC curve of changes in Fe predicting mortality. (**B**) ROC curve of Fe predicting SOFA at ICU admission. (**C**) ROC curve of Zn predicting APACHE at ICU admission. (**D**) ROC curve of Zn predicting SOFA on the 3rd day of ICU stay.

**Table 1 metabolites-13-00931-t001:** Biochemical, inflammatory, mineral, and metallothionein values of critically ill patients with COVID-19 on the 1st and 3rd days of admission to the ICU.

*n* = 86	References Values	1st Day (Mean (SD))	3rd Day (Mean (SD))	MeanDifferences	*p*-Value	BH-P
**Biochemical and Inflammatory Parameters**						
Albumin (g/dL)	3.5–5.2	3.2 (0.5)	3.3 (3.0)	+0.1	0.923	0.923
Ferritin (ng/mL)	20.0–275.0	1641.2 (1130.6)	1768.7 (1287.6)	+127.5	0.213	0.280
Transferrin (mg/dL)	200.0–360.0	146.1 (25.5)	161.6 (41.5)	+15.5	**0.036**	0.064
TSI (%)	17.1–30.6	46.1 (34.5)	38.3 (26.9)	−7.9	0.183	0.269
Fibrinogen (mg/dL)	200.0–350.0	651.1 (211.1)	556.4 (189.9)	−94.6	**0.001**	**0.025**
D-dimer (ng/mL)	0.0–500.0	1291.0 (1246.9)	2049.6 (1842.5)	+751.3	**0.001**	**0.012**
CRP (mg/L)	0.0–5.0	120.6 (86.8)	75.5 (70.6)	−45.2	**0.001**	**0.008**
GOT (U/L)	5–40	42.2 (29.7)	37.5 (32.7)	−4.7	0.238	0.297
GPT (U/L)	0–55	48.5 (46.5)	62.1 (73.7)	+13.6	**0.025**	**0.048**
GGT (U/L)	1–55	99.8 (108.6)	142.3 (198.3)	+42.5	**0.008**	**0.020**
LDH (U/L)	0–248	519.6 (19.1)	463.1 (187.8)	−56.5	**0.008**	**0.018**
Hb (g/dL)	11.0–17.0	13.1 (2.0)	12.4 (2.1)	−0.7	**0.001**	**0.006**
Hematocrit (%)	30.0–50.0	38.4 (5.7)	36.7 (5.7)	−1.7	**0.001**	**0.005**
Leukocytes (×10^3^/µL)	3.5–10.5	11.6 (6.2)	10.5 (5.4)	−1.1	**0.046**	0.076
Lymphocytes (%)	20.0–44.0	7.2 (4.4)	9.7 (8.1)	+2.5	**0.005**	**0.017**
Neutrophils (%)	42.0–77.0	88.4 (5.6)	82.3 (13.8)	−6.1	**0.001**	**0.004**
Platelets (×10^3^/µL)	120.0–450.0	232.5 (89.6)	256.6 (104.5)	+24.1	**0.006**	**0.016**
TNT (ng/L)	<14.0	19.7 (41.5)	12.9 (28.6)	−6.8	0.151	0.235
APTT (s)	26.0–37.0	28.9 (3.9)	29.2 (4.2)	−0.3	0.655	0.711
INR	0.80–1.2	1.1 (0.3)	1.0 (0.2)	−0.1	0.367	0.436
**Minerals**						
Iron (mg/L)	0.6–1.70	1.7 (0.9)	1.5 (0.9)	−0.2	0.197	0.273
Zinc (mg/L)	0.7–1.10	0.9 (0.3)	0.8 (0.4)	−0.1	0.607	0.689
Copper (µg/L)	0.6–1.40	0.6 (0.4)	0.5 (0.3)	−0.1	**0.005**	**0.015**
Manganese (µg/L)	0.4–0.85	0.5 (0.1)	0.4 (0.1)	−0.1	0.704	0.733
**Metallothioneins**						
MTs (µmol/L)	-	0.3 (0.1)	0.2 (0.1)	−0.1	**0.022**	**0.045**

*n* = 86. Data are expressed as mean (standard deviation) unless otherwise stated. Paired Student’s *t*-test was used to compare the evolution of the variables between 1st and 3rd days of the ICU stay. Mean differences represent the difference between the value of the 3rd and 1st days of ICU stay. Significant *p*-Values are represented in bold. *p*-Values after applying the Benjamini–Hochberg (BH) procedure for controlling False Discovery Rate (FDR) are presented. Statistical significance = *p* < 0.05. Abbreviations: APTT = Activated Partial Thromboplastin Time; CRP = C-Reactive Protein; Cu = copper; Fe = iron; GOT = Glutamic Oxaloacetic Transaminase; GPT = Glutamic Pyruvic Transaminase; GGT = Gamma-Glutamyl Transferase; Hb = Hemoglobin; INR = International Normalized Ratio; LDH = Lactate Dehydrogenase; Mn = manganese; MTs = metallothioneins; SD = standard deviation; TNT = Troponin T; TSI = Transferrin Saturation Index; Zn = zinc.

**Table 2 metabolites-13-00931-t002:** Changes in clinical, biochemical, and inflammatory parameters based on the median changes in minerals and MTs occurring during the ICU stay.

*n* = 86 Δ Change	Δ Fe			Δ Zn			Δ Cu			Δ Mn			Δ MTs		
	Δ	*p*-Value	BH-P	Δ	*p*-Value	BH-P	Δ	*p*-Value	BH-P	Δ	*p*-Value	BH-P	Δ	*p*-Value	BH-P
**Clinical parameters**															
SOFA	+0.05	0.917	0.917	+0.45	0.576	0.706	+0.06	0.908	1.000	+0.88	0.170	0.765	+0.08	0.895	0.966
MAP	+5.50	0.506	0.803	–2.40	0.893	0.893	+16.6	0.095	0.427	+12.8	0.182	0.702	+3.62	0.658	1.000
HR	−25.1	**0.006**	0.162	–6.15	0.592	0.694	+2.88	0.807	1.000	−11.5	0.295	0.885	+3.73	0.677	1.000
BR	−0.48	0.862	0.930	–1.43	0.329	1.000	+3.23	0.253	0.569	−2.09	0.521	0.879	+2.15	0.509	1.000
FiO_2_	+0.01	0.806	0.906	–0.03	0.456	0.769	−0.04	0.439	0.911	+0.05	0.480	0.925	−0.05	0.431	1.000
PaO_2_/FiO_2_	+23.1	0.618	0.794	–0.12	0.415	0.800	+21.6	0.679	1.000	+43.1	0.499	0.898	+74.1	0.151	1.000
PEEP	+0.93	0.266	1.000	–1.23	0.146	1.000	−1.10	0.242	0.594	−0.87	0.396	0.891	+0.81	0.322	1.000
**Biochemical and inflammatory parameters**															
Albumin	+0.75	0.325	0.731	−2.25	0.363	0.980	−0.04	0.617	1.000	+1.95	**0.044**	0.594	−0.81	0.359	1.000
Ferritin	+177.7	0.392	0.814	−71.5	0.884	0.918	+15.5	0.948	0.948	−92.8	0.720	1.000	−117.4	0.613	1.000
Transferrin	−23.4	0.091	0.491	+26.4	0.214	0.963	−1.19	0.944	0.980	−5.11	0.764	0.982	+1.25	0.943	0.943
TSI	+12.1	0.301	0.738	−17.5	0.347	1.000	+25.5	**0.031**	0.161	+19.0	0.153	1.000	−5.02	0.726	1.000
Fibrinogen	+27.2	0.527	0.790	+58.4	0.427	0.768	+148.5	**0.001**	**0.027**	+54.1	0.319	0.861	+15.3	0.737	0.947
D-dimer	−1315.0	0.294	0.793	+1797.0	0.484	0.726	+504.6	0.723	1.000	−1526.2	0.340	0.834	+265.7	0.855	0.961
CRP	−13.8	0.438	0.788	+18.3	0.605	0.680	+44.0	**0.017**	0.229	−45.4	**0.038**	1.000	+6.62	0.734	0.990
GOT	−13.9	0.090	0.607	+9.71	0.402	0.904	−2.09	0.812	0.996	−15.8	0.115	1.000	−2.01	0.832	1.000
GPT	−22.1	0.073	0.985	+11.7	0.544	0.699	−19.9	0.121	0.408	+2.51	0.866	1.000	−8.50	0.522	1.000
GGT	+35.2	0.284	0.958	+25.4	0.471	0.748	+43.4	0.229	0.618	+87.1	0.211	0.712	+24.6	0.498	1.000
LDH	+31.6	0.464	0.783	−50.9	0.371	0.910	+78.0	0.104	0.401	+18.0	0.746	1.000	+68.2	0.151	1.000
Hb	−0.12	0.615	0.830	+0.28	0.524	0.744	+0.18	0.505	0.973	+0.06	0.843	1.000	+0.30	0.270	1.000
Haematocrit	−0.61	0.432	0.833	+0.90	0.527	0.711	+0.46	0.588	1.000	+0.34	0.714	1.000	+0.62	0.463	1.000
Leukocytes	+0.38	0.719	0.844	+2.63	0.113	1.000	+1.41	0.221	0.663	+0.20	0.876	0.985	+0.14	0.898	0.932
Lymphocytes	−0.85	0.639	0.784	−0.67	0.880	0.950	−4.57	**0.023**	0.207	+0.95	0.671	1.000	+1.74	0.364	1.000
Neutrophils	+1.63	0.605	0.859	−6.33	0.407	0.845	+6.51	**0.030**	0.202	−0.46	0.890	0.961	−3.48	0.243	1.000
Platelets	−2.33	0.897	0.931	+54.9	0.085	1.000	−6.65	0.716	1.000	+0.53	0.979	0.979	−16.1	0.416	1.000
TNT	+13.5	0.290	0.870	+9.25	0.208	1.000	−1.12	0.936	1.000	+24.2	0.161	0.869	−11.1	0.444	1.000
APTT	−1.74	0.098	0.441	+1.85	0.192	1.000	−0.20	0.849	0.996	−1.07	0.401	0.832	−0.62	0.585	1.000
INR	−0.12	0.075	0.675	−0.14	0.258	0.995	+0.02	0.811	1.000	+0.01	0.942	0.932	−0.01	0.841	0.987

*n* = 86. Data are expressed as mean (standard deviation) unless otherwise stated. Δ represents the difference between the value on the 3rd and the 1st days of the ICU stay. Unpaired Student’s *t*-test was used to compare the changes in clinical, biochemical, and inflammatory parameters of the study based on the median changes in the levels of minerals and MTs. Significant *p*-Values are represented in bold. *p*-Values after applying the Benjamini–Hochberg (BH) procedure for controlling the False Discovery Rate (FDR) are presented. Statistical significance = *p* < 0.05. Abbreviations: APTT = Activated Partial Thromboplastin Time; BR = Breathing Rate; CRP = C-Reactive Protein; Cu = copper; Fe = iron; GOT = Glutamic Oxaloacetic Transaminase; GPT = Glutamic Pyruvic Transaminase; GGT = Gamma-Glutamyl Transferase; Hb = Hemoglobin; HR = Heart Rate; INR = International Normalized Ratio; LDH = Lactate Dehydrogenase; MAP = Mean Arterial Pressure; MTs = Metallothioneins; PaO_2_/FiO_2_ = Partial Oxygen arterial pressure/Fraction of Inspired Oxygen; PEEP = Positive End-Expiratory Pressure; SOFA = Sequential Organ Failure Assessment. TNT = Troponin T; TSI = Transferrin Saturation Index; Zn = zinc.

**Table 3 metabolites-13-00931-t003:** Matrix for correlation coefficients (r) showing the simple linear relationship between minerals and MTs with clinical, biochemical, inflammatory, mineral, and MT parameters of the study.

*n* = 86 Δ Change	Δ Fe	Δ Zn	Δ Cu	Δ Mn	Δ MTs
**Clinical parameters**					
SOFA	−0.076	+0.026	−0.127	+0.209	+0.042
MAP	+0.087	−0.377	+0.226	+0.159	+0.300
HR	−0.258	−0.189	−0.079	+0.073	+0.146
BR	−0.229	**–0.815** *	+0.251	−0.366	−0.133
FiO_2_	0.058	−0.179	−0.220	+0.390	+0.142
PaO_2_/FiO_2_	−0.166	**+0.963** *	+0.262	−0.653	+0.113
PEEP	−0.002	−0.295	−0.083	+0.288	−0.095
**Biochemical and inflammatory parameters**					
Albumin	+0.060	−0.059	–0.068	+0.043	**−0.328** *
Ferritin	+0.054	+0.223	+0.069	+0.021	+0.030
Transferrin	–0.163	+0.359	–0.050	+0.029	+0.025
TSI	+0.066	−0.138	+0.404	+0.232	−0.230
Fibrinogen	+0.131	−0.148	**+0.447** **^,a^	**+0.250** *	+0.077
D-dimer	+0.015	+0.185	+0.081	−0.066	+0.033
CRP	–0.015	−0.264	**+0.276** *	−0.079	+0.053
GOT	+0.029	−0.114	−0.086	**−0.221** *	+0.030
GPT	+0.031	+0.115	−0.195	−0.164	−0.054
GGT	+0.114	+0.215	+0.063	+0.113	+0.001
LDH	+0.112	−0.210	+0.205	+0.145	+0.103
Hb	–0.035	+0.219	+0.160	−0.111	+0.083
Hematocrit	–0.064	+0.313	+0.122	−0.151	0.051
Leukocytes	+0.153	+0.070	+0.129	+0.078	−0.194
Lymphocytes	–0.139	−0.031	−0.204	+0.012	+0.060
Neutrophils	–0.003	−0.183	+0.173	+0.011	−0.076
Platelets	+0.117	**+0.494** *	+0.076	+0.068	−0.193
TNT	+0.001	+0.156	+0.019	+0.298	−0.022
APTT	–0.174	−0.065	+0.006	−0.117	−0.064
INR	–0.147	**−0.480** *	−0.199	−0.154	+0.040
**Minerals**					
Fe	-	-	-	**-**	+0.039
Zn	−0.262	-	-	-	−0.152
Cu	**+0.282 ***	−0.330	-	-	−0.022
Mn	**+0.266 ***	+0.197	+0.059	-	**−0.255** *
**Metallothioneins**					
MTs	-	-	-	-	-

*n* = 86. Pearson’s correlation was used to compare the simple linear relationship between changes in minerals and MTs and clinical, biochemical, inflammatory, mineral, and MT values. Δ represents the difference between the value of the 3rd and the 1st days of the ICU stay. Significant *p*-Values are represented in bold. Matrix correlations are presented as correlation coefficients (r). *p*-Values after applying the Benjamini–Hochberg (BH) procedure for controlling False Discovery Rate (FDR) are presented. Statistical significance before adjustment = * *p* < 0.05 and ** *p* < 0.001; Statistical significance after adjustment = ^a^
*p* < 0.001. Abbreviations: APTT = Activated Partial Thromboplastin Time; BR = Breathing Rate; CRP = C-Reactive Protein; Cu = copper; Fe = iron; GOT = Glutamic Oxaloacetic Transaminase; GPT = Glutamic Pyruvic Transaminase; GGT = Gamma-Glutamyl Transferase; Hb = Hemoglobin; HR = Heart Rate; INR = International Normalized Ratio; LDH = Lactate Dehydrogenase; MAP = Mean Arterial Pressure; MTs = Metallothioneins; PaO_2_/FiO_2_ = Partial Oxygen arterial pressure/Fraction of Inspired Oxygen; PEEP = Positive End-Expiratory Pressure; SOFA = Sequential Organ Failure Assessment. TNT = Troponin T; TSI = Transferrin Saturation Index; Zn = zinc.

## Data Availability

Data will be shared upon reasonable request by the corresponding authors: Héctor Vázquez-Lorente and Lourdes Herrera-Quintana.

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
