# Peer review of "Evolution of Status of Trace Elements and Metallothioneins in Patients with COVID-19: Relationship with Clinical, Biochemical, and Inflammatory Parameters"

_metabolites, 2023, doi:10.3390/metabo13080931_

Round 1
Reviewer 1 Report
Article: Evolution of trace elements and metallothioneins status in patients with COVID-19: relationship with clinical, biochemical, and inflammatory parameters.
Article presents a study cohort comprised of COVID patients admitted to the ICU. Basic clinical/biochemical/inflammatory parameters were assessed including mineral and metallothionein levels. Patients were only assessed at 1 to 3 days ICU stay, and correlations were conducted between measured parameters and ultimate patient mortality and severity. Authors claim Fe and Zn displayed a predictive value for mortality and severity following admittance to the ICU with COVID.
Questions:
Line 82: Methods. Why were patients only assessed up to 3 days ICU stay?
Line 188 Supp Table 1: ICU stay. What was the average ICU stay for patients surviving vs not surviving?
Line 188 Supp Table 1: it may be interesting to separate patients into two groups. Surviving vs not surviving, at least for the Supplementary table.
Line 188 Supp Table 1: Would not BR be influenced by MV?
Line 196: Table 1. You should be able to assess significance between Reference values and 1 day values? Reference values should have a mean ± SD making this possible?
Line 196: Table 1. You have in effect three data groups, Reference values, 1 day, and 3 days. Could you use ANOVA to compare data sets?
Line 204: Figure 1. "Paired Student’s t-test was used to compare the evolution of the variables." What does this mean? Is this the correct usage of students-t? How does this differ from that already shown in table 1?
Line 211: Table 2. Confused as to how this data was grouped/analysed. Table has columns for each Δ mineral and Δ MT. Then a list of parameters with P-Values within each column. Please better explain how these values were grouped and analysed. How is this data different that correlation data shown in table 3?
Line 223: Table 3. why was r used instead of r2
Comments:
Line 34: Introduction, paragraph 1. Excessive use of dashes in text (i.e. "—"), not typical in normal English writing.
Line 188: Supp Table 1. Should be formatted to be all on same page
Line 196: Table 1. Should be formatted to be all on same page
Line 184: Result section is simply a collection of tables and figures, there is zero text describing the results from this study. A small amount of results text appears within the discussion section and this should be moved to its correct location and greatly expanded upon. If presenting a joint results/discussion this should be made clear in section headings
Suggestions:
Line 24: suggest change "differential pulse voltammetric" to "differential pulse voltammetry"
Line 25: suggest change "Levels of Cu and MTs decreased (all P≤0.046) after 3 days" to "Levels of Cu and MTs were significantly decreased after 3 days (P < 0.05)."
Line 26: suggest change "Changes in Fe were directly related to changes in Cu and Mn (all r≥0.266; P≤0.019). In contrast, changes in MTs were inversely related to changes in Mn and albumin (all r≥–0.255; P≤0.039)" to
"Changes in Fe were found to directly correlate with changes in Cu and Mn (r≥0.266; P<0.05). In contrast, changes in MTs were inversely correlated to changes in Mn and albumin (r≥–0.255; P≤0.05) "
Line 28: suggest change "The present study indicated an apparent risk of trace element deficiency" to "The present study suggests trace element deficiency may be a risk factor during early ICU treatment of COVID-19."
Line 47: "may be an interesting alternative." Citation(s) needed (e.g. PMID: 32340216)
Line 63: suggest change "Importantly, metal ion concentrations in the body" to "Metal ion concentrations in the body"
Line 75: suggest change "subsequent effects on clinical and inflammatory parameters have not been investigated, at least in the context of COVID-19 disease. " to "subsequent effects on clinical and inflammatory parameters during COVID-19 has not been investigated. "
Line 234: Figure 2. Please label ROC curves indicating mineral and parameter on each curve/axis.
Overall quality of English is good except for:
Line 34: Introduction, paragraph 1. Excessive use of dashes in text (i.e. "—"), not typical in normal English writing.
Author Response
We are grateful for the effort and suggestions. The attached word document contains a list of each and every one of the responses to the proposed comments.

Reviewer 2 Report
The authors present an analysis of trace element and metallothionein levels in patients with COVID-19, by ICP-MS. The manuscript is well-written, in an area that is much less extensively researched than other blood-based analyses such as 'omics. The only major issue relates to the use of false discovery correction in the analysis - it is not clear how the authors have dealt with this. Some additional minor comments and suggestions are set out below.
1. Introduction
[line 48] "Thus, some current therapies in clinical settings consist of immuno- modulatory agents – including micronutrients to fight against oxidative stress and modulate inflammation – [6]."
Please remove the trailing hyphen.
[line 78] "in critical care patients with COVID-19 at ICU admission and after 3 days of ICU stay. We hypothesized that trace elements could be directly related to MTs levels – their deficiencies worsening 80 during the ICU stay –."
This sentence is not correct, either remove the trailing hyphen at the end or complete the sentence (also see previous comment).
2.1. Subjects and study design
Did the authors have information on medication regime? The recruitment period includes the time when dexamethasone was used - this can impact metabolomic profiles (see https://doi.org/10.3390/ijms232012079), and so might influence other biomarkers including those analysed here. If the authors do have the data, it might be useful to analyse whether the glucocorticoid treated group saw a different response on Day 3. If they did not, then treatment regime would not have been a confounder.
Other variables (including fasting) appear well-controlled for.
2.4. Statistical analysis
[Lines 176-177] "The predictive value of Fe and Zn levels for mortality and severity at baseline and follow-up was evaluated using the Receiver Operating Curve (ROC)."
Some more detail on what approach was used for the ROC would be helpful; if each biomarker was used independently was this a simple analysis of cut-off point determining sensitivity and specificity?
3. Results
Tables 1, 2 and 3 are very detailed. It would be helpful to the reader to have a Figure with boxplots for the key identified markers (Cu, Fe etc). Boxplots are much more informative than simple mean and standard deviations.
Also on Tables 1, 2 and 3. You are measuring a large number of variables. When analysing multiple variables, false discovery rate correction methods are appropriate to use. It is not clear to me whether the authors have used e.g. Benjamin-Hochberg or other methods to appropriately calculate the p-values. See Broadhurst et al: https://doi.org/10.1007/s11306-006-0037-z
Figures 1 and 2: The editor will also advise on house style, but in general, boxes around figures are not necessary, and do not look very pleasing to the eye. Not a criticism, just a suggestion.
4. Discussion
Some additional discussion about associations of these biomarkers with related conditions might be helpful, for example see Livesey et al: http://dx.doi.org/10.1136/thoraxjnl-2011-201076
Are these marker changes specific to COVID-19, or are they reflective of the symptoms, such as inflammation or thrombosis, and might be seen in other conditions with the same symptoms?
[lines 315 to 317] "In contrast, Hackler et al. [37] found elevated serum 315 Cu levels at hospital admission in the surviving patients compared with non-survivors 316 and with the reference range – no patient showing Cu levels below the reference range –. "
This sentence appears to be incomplete
[line 354] As a layperson in this field, it was not clear to me why non-COVID controls in ICU were not measured alongside COVID ICU patients. Is this a meaningful limitation? If not, why not?
[line 357] Hospital does not need a capital letter here
Aside from some minor comments detailed above (especially trailing hyphens), the manuscript is written to a good standard.
Author Response

(The authors gave the same response as above.)

Round 2
Reviewer 2 Report
The authors have not appropriately dealt with the major point I raised in the first review.
"Response 6: Dear reviewer, when we obtain p-values, we further check that the reported significance is correct by using tools such as visual checking of histograms, box plots and Q-Q plots. We have been thoroughly analyzing the proposed article and the Benjamin-Hochberg method afterwards and have assumed a false discovery rate of 5%. Considering that the significant p-values were very low, as well as the probability of exclusion was around one variable, we considered it unnecessary to exclude any variable. However, we are very grateful for this statistical approach provided, because in future studies with a larger number of variables, we may find some variables under false discovery rate."
False-discovery correction is essential when reporting p-values for multiple hypothesis testing. Please report FDR adjusted values in your tables. If the authors are having difficulty calculating them, uploading the data to the website MetaboAnalyst as a .csv with the values for the samples split across two populations will allow a variety of statistical tests on a matrix of n samples versus p features, and the authors can obtain FDR corrected p-values there. Alternatively, such p-values can be calculated in any statistical package.
Round 3
Reviewer 2 Report
Thank you for including FDR adjusted p-values. I believe the manuscript is interesting and should be published.